# Evidence of Long-Distance Aerial Convection of Variola Virus and Implications for Disease Control

**DOI:** 10.3390/v12010033

**Published:** 2019-12-27

**Authors:** Chandini Raina MacIntyre, Arpita Das, Xin Chen, Charitha De Silva, Con Doolan

**Affiliations:** 1Biosecurity Program, The Kirby Institute, UNSW Medicine, University of New South Wales, Sydney, NSW 2052, Australia; rainam@protonmail.com (C.R.M.); arpitadas1908@gmail.com (A.D.); 2College of Health Solutions, Arizona State University, Phoenix, AZ 85004, USA; 3College of Public Service and Community Solutions, Arizona State University, Phoenix, AZ 85004, USA; 4School of Mechanical and Manufacturing Engineering, University of New South Wales, Sydney, NSW 2052, Australia; charithads@protonmail.com (C.D.S.); cdoolan@protonmail.com (C.D.)

**Keywords:** smallpox, variola virus, aerial convection, airborne, systematic review

## Abstract

Two distinct phenomena of airborne transmission of variola virus (smallpox) were described in the pre-eradication era—direct respiratory transmission, and a unique phenomenon of transmission over greater distances, referred to as “aerial convection”. We conducted an analysis of data obtained from a systematic review following the PRISMA criteria, on the long-distance transmission of smallpox. Of 8179 studies screened, 22 studies of 17 outbreaks were identified—12 had conclusive evidence of aerial convection and five had partially conclusive evidence. Aerial convection was first documented in 1881 in England, when smallpox incidence had waned substantially following mass vaccination, making unusual transmissions noticeable. National policy at the time stipulated spatial separation of smallpox hospitals from other buildings and communities. The evidence supports the transmission of smallpox through aerial convection at distances ranging from 0.5 to 1 mile, and one instance of 15 km related to bioweapons testing. Other explanations are also possible, such as missed chains of transmission, fomites or secondary aerosolization from contaminated material such as bedding. The window of observation of aerial convection was within the 100 years prior to eradication. Aerial convection appears unique to the variola virus and is not considered in current hospital infection control protocols. Understanding potential aerial convection of variola should be an important consideration in planning for smallpox treatment facilities and protecting potential contacts and surrounding communities.

## 1. Introduction

Smallpox was a widespread disease in humans caused by the variola virus, which is a member of the poxvirus family [1]. Smallpox was declared eradicated by the World Health Organization (WHO) in 1980, following successful vaccination campaigns and other favorable conditions [2]. However, smallpox poses a public health threat of re-emergence due to the absence of mass vaccination, waning immunity and advances in synthetic biology, which make synthesis of variola possible [3,4]. Smallpox transmission occurs from person to person, primarily through respiratory droplets but can also transmitted through contact with infected clothing and bedding [5]. The R0 is estimated to be around 5 [6]. Whilst most pathogens have a dominant mode of transmission, they usually have more than one mode of transmission, and quantifying the relative contributions of different modes of transmission in the spread of an infection is difficult [7]. Two distinct phenomena of airborne transmission are described with variola—firstly, direct person-to-person transmission by the airborne respiratory route, and secondly, transmission over greater distances, sometimes over a mile, referred to as “aerial convection” [8].

It is believed that variola virus can transmit through the airborne, aerosol and contact routes, and some cases of indirect spread via fine particle aerosols or fomites have been reported [9,10,11,12]. The risk of person-to-person airborne infection is a function of the virus concentration in respiratory fluid, the expiratory event rate, the size and volume distribution of the particles emitted per expiratory event, the receptor’s breathing rate, exposure duration, and the receptor’s location in the room relative to the source case [12]. There have been various experimental studies on animals, which confirm that infection can be produced by a single plaque-forming unit (PFU) of virus carried in respirable particles—even a submicrometric aerosol of variola can cause infection in animals [12,13,14,15].

Smallpox can be transmitted through aerosols and the airborne route, as evidenced by air-sampling techniques together with culture and molecular detection methods [16]. There is also evidence showing the association between ventilation and the control of airflow directions in buildings and the transmission of infectious diseases [17]. Airborne transmission accounts for at least 10% of all nosocomial infections, based on patient-based surveillance systems and environmental sampling techniques in hospital settings [16,18].

Smallpox can survive in the environment under certain conditions and contaminated particles may be still isolated from the air in the later stages of the disease corresponding to a smaller number of lesions [19]. Thomas (1974) recovered the smallpox virus from an isolation unit using an adhesive surface air-sampling technique in the presence of very low aerosol concentrations [20]. A review of the literature suggests that the role of airborne transmission may have been underestimated in many instances [21].

In addition to direct respiratory transmission from person to person within 1–2 m of spatial separation, a more distant transmission has been described. Whilst it is well-established that airborne infection can occur [8,19], the spread of smallpox by means of “aerial convection” is less well understood. Aerial convection refers to transmission over a substantial distance, (greater than expected during direct person to person respiratory transmission of 1–2 metres and possibly aided by wind or air currents) a concept accepted by many epidemiologists. In recognition of this, the Ministry of Health regulations in Britain in the 1940s stipulated that smallpox hospitals should be “at least a quarter of a mile from another institution or a population of 200, and at least a mile from a population of 600” [8].

Aerosol transmission can occur over short distances or long distances, and the transmission is primarily governed by air flows driven by pressure differences generated by ventilation systems, open windows and doors, movement of people or temperature differences [16]. Aerosolised particles have the potential to remain suspended in the air for hours and can expose a larger number of susceptible individuals to potential infections at a greater distance from the source [21]. Our understanding of respiratory transmission and measurement methods have improved substantially since eradication [22], which makes it timely to review the evidence. Now 40 years since clinicians and public health agencies have managed smallpox, there is a need to review the pre-eradication human evidence around smallpox transmission, especially the role of aerial convection, for which there is little awareness among contemporary clinicians. This is essential to inform disease control strategies and health care worker occupational safety in preparedness planning.

The aim of this paper was to study the data and epidemiology of transmission of variola by aerial convection, and examine the hypothesis that aerial convection was only observed when the incidence of smallpox is sufficiently low to exclude other sources of infection.

## 2. Materials and Methods

Given the phenomenon of aerial convection was observed in the last 100 years of smallpox endemicity in the world, we examined the incidence of smallpox using the best available data, in relation to the period of observation of aerial convection. We used the systematic review methodology to identify data and evidence for aerial convection, and then analysed incidence data, documented distances of transmission, and the period of observation of aerial convection in relation to the epidemiology of smallpox deaths.

### 2.1. Smallpox Data Analysis

In order to clearly demonstrate the window during which aerial convection was observed relative to the epidemiology of the disease, we created the timeline of all identified observations of aerial convection that was plotted over the number of smallpox deaths using available data in London from 1700–1980. London was selected because there was better data for deaths (than cases) than global data or other countries, and the largest number of observations of aerial convection were from England.

We collected the data of the annual number of deaths from smallpox in London from 1700 to 1902 [23]. Due to missing data on deaths in London between 1903 and 1958, we collected the available data to estimate data for London, including (1) the number of deaths from smallpox in England and Wales from 1911 to 1919 [24], (2) population in London, England and Wales in 1911 (26), (3) number of cases and case fatality rate in the UK from 1920–1958 [25], and (4) the number of deaths in London between 1959 and 1980 [26,27,28].

We estimated that the annual deaths in London were 13% of deaths in England and Wales between 1911 and 1919, according to the population in London (4.52 million) taking 13% of the population in England and Wales (36.07 million) in 1911 [29]. Using the case fatality rate of 30% [30], we calculated the annual number of deaths in the UK from 1920 to 1958 based on the yearly reported cases [31]. The annual deaths in London were 17% of deaths in the UK between 1920 and 1958, according to the population in London (7.39 million) [32] taking 17% of the UK population (43.90 million) in 1921 [33]. The deaths in London between 1958 and 1980 were extracted from documented smallpox outbreaks in the UK [26,27,28]. These different data sources were used to create a timeline of smallpox deaths in London from 1700–1980. Then we plotted all identified observations of aerial convection over this timeline, to show the relationship of this observation period of aerial convection to the epidemiology of smallpox and the waning incidence of smallpox.

### 2.2. Search Strategy

A systematic review was conducted on finding evidence for aerial transmission of smallpox. Our review focused on identifying smallpox outbreaks occurring in humans and studying the reported transmission pattern to find evidence of aerial convection spread. Specifically, we looked for cases where people acquired smallpox in the vicinity of, but at a distance of >2 m from a known smallpox case. We searched five databases: Medline (1946 to present), Embase (1974 to present), Scopus (1960 to present), Web of Science (1898 to present), Global Health (1910 to present). Results were limited to peer-reviewed publications in English. Search terms used were “smallpox” OR “variola” AND “transmission” AND “outbreak”. We also reviewed the bibliography of retrieved articles to identify other references that might not otherwise have been identified. Secondary searches were conducted which included the cross-references from various relevant papers. Author AD independently screened each title and abstract in the search result and in case of uncertainty, the author (CRM) was consulted. A full-text evaluation was conducted by three review authors (CRM, AD and XC). The results from our review are presented in accordance with the PRISMA guidelines, which is the accepted standard for a systematic review (Figure 1) [34].

### 2.3. Selection Criteria and Inclusion

We focused on the aerial transmissibility of smallpox in outbreaks and on the evidence of transmission route from analysis of relevant outbreaks. Eligible studies had to fulfil the following criteria: (1) Peer-reviewed journal articles, (2) published in English, (3) primary focus on smallpox outbreaks in humans, (4) outbreak case studies showing evidence of transmission at distances >2 m in the absence of known smallpox transmission in the surrounding community, (5) studies on variola major or variola minor.

### 2.4. Exclusion Criteria

Following the full-text assessment, we excluded (1) an outbreak or case of smallpox acquired where the acquired case was in close proximity (within a distance of 2 m) of a known case of smallpox, enabling direct respiratory transmission by droplets or aerosols, or through physical contact, or where the outbreak occurred in the presence of known smallpox transmission in the surrounding community, making transmission too widespread to observe long-distance transmission. (2) Laboratory experiments and studies unrelated to transmission route of smallpox, (3) non-human studies, (4) mathematical modelling studies, (5) studies focussing on smallpox eradication or vaccination, and (6) studies focusing on vaccinia virus.

### 2.5. Review

Firstly, we read all titles and abstracts of the studies identified through the search and after removing duplicates, we included relevant, eligible papers for full-text assessment studies. While considering the contributory role of airborne transmission of smallpox, we assumed that fulfillment of the following conditions indicated evidentiary support [17].

An outbreak or case of smallpox in a setting that occurred due to transmission of infectious particles from one location to another spatially separate location, farther than possible through direct person to person respiratory transmission (>2 m) and without evidence of other community transmissions that could have explained incident infections. The term “aerial convection” (used in the pre-eradication era) was used for distant transmission.

All the review authors were asked to consider these criteria while rating the findings of each study as “Conclusive” if aerial convection was the only explanation for one or more transmissions or “Partially conclusive” if aerial convection was one explanation for one or more transmissions, but other explanations were also possible [35].

## 3. Results

We identified 22 studies meeting the inclusion criteria, which described 17 different outbreaks. The outbreaks with evidence of transmission route from the selected studies are summarized in Table 1. Eight outbreaks involved very long-range transmission beyond a single building or location and nine involved transmission within a building that could not be explained by direct person to person contact. This included two outbreaks where transmission occurred vertically from one floor to another floor of a building.

Figure 2 summarises the distances of aerial convection of smallpox in different outbreaks described in Table 1, where distance was quantified. This ranged from half a mile (0.8 km) to over 9 miles (15 km) in the case of the Aralsk outbreak. Most were between 0.5–1 mile (0.8–1.6 km).

Figure 3 shows the epidemiology of smallpox cases in London and the window during which aerial convection was observed, which was in a period of the very low incidence of infection, from 1881 to 1978, two years before eradication.

The outbreaks we studied showed two different patterns of transmission. One was of very long-range transmission over distances of a half-mile to a mile. This comprised single source cases and a secondary case occurring at a long horizontal distance, or clustering of cases in the community within a radius around a smallpox hospital (building or ship). The second was transmission within a building (often vertically from one floor to another) or between adjacent buildings in the absence of relevant contact with an infected case. The Birmingham case [27,28], the Meschede outbreak [9,10,11,17,41,42,49] and the New York outbreak [39] showed transmission from one floor of a building to another, suggesting air currents can carry virus either through the air conditioning systems or through open windows. One study supporting aerial convection of variola is the controlled experimental work conducted in the Meschede, Germany outbreak via smoke flow visualisation [9,10,49]. These results clearly revealed that the airflow patterns matched the location of cases within buildings. At the time, however, such studies were limited, and emphasis was placed on trying to explain transmission by other modes of transmission.

The second pattern was long-distance horizontal transmission from one site to another of a mile or more, as seen in Fulham [8,36], Salonika [37], Gravesend [37], and Purfleet [8], among others. The Fulham data are particularly convincing because of the painstaking collection of data and mapping of smallpox cases in proximity to hospitals, and the fact that the initial observations in Fulham were reproduced around the country. [8,36] The observations from the smallpox ships in the Thames and recurrent epidemics in the nearest communities to shore were also supportive. An outlier case was that of the Lev Berg ship in the Aral sea. In this instance, it is likely that the official reports (which state the index case got off the ship at several stops) was an obfuscation of the truth, and that the comments of Dr Pyotr Burgasov after the collapse of the Soviet Union (that a 400 g smallpox “bomb” was exploded on Vozrozhdeniye Island) were closer to the truth [50]. The Lev Berg sailed 15 km off the coast of Vozrozhdeniye Island, which was a known Soviet biological weapon testing site. It seems likely this was the source of infection, and possible that a weaponised attack may disperse the virus at least 15 km.

## 4. Discussion

Understanding smallpox transmission is crucial for preparedness planning and can inform control of a re-emergent epidemic of smallpox. It was accepted that variola was most effectively transmitted by the respiratory route, and it was formerly called a “preferentially” airborne infectious disease” [51]. Variola virus has been recovered in airborne droplets from air sampling, supporting aerosol transmission [20]. The observation of aerial convection in a number of outbreaks further supports airborne transmission. Aerial convection, however, was more controversial and is not a concept that is currently in the corporate memory or included in hospital infection control protocols. We found supportive evidence of aerial convection from 12 out of 17 outbreaks, and a further 5 outbreaks which were partially conclusive. The examples of transmission from one floor to another or one building to another, presumably by air currents, are more easily explained, as distances were shorter and supported by the smoke experiments at Meschede, Germany [9,10,49]. In the last documented case of smallpox in the world, the Birmingham case, we can be fairly certain the patient was infected from a virus in the laboratory. Case ascertainment was high at that time, the location was the UK, which had long since eliminated smallpox, so it is likely the source of infection was an aerosolised virus through air-conditioning dust or an open window. More recently, the transmission of SARS in the Amoy Gardens building, where aerosolised faecal material spread from floor to floor through plumbing and open bathroom grates, but also from open windows to adjacent buildings, demonstrated that air currents can carry virus particles from one building to another [52]. However, in over half of the outbreaks we reviewed, there were reports of infections from a single index patient that were between a quarter to one mile apart in the absence of other smallpox cases in the community. At such distances, other modes of transmission are largely infeasible, although secondary aerosolization from contaminated clothing or bedlinen carried from the patient room to the community is possible. Another possibility in the apparently long-range transmissions is that these were exposed to missed mild or vaccine-modified cases. However, it should be noted in the case in Greece, that the secondary case occurred within the incubation period of the index case being symptomatic [37].

The theory of aerial convection is biologically plausible. Several studies in humans and animals have shown that virus concentration is higher in the lower respiratory tract than the upper respiratory tract and that the infectious dose is very low, consistent with smaller airborne infectious particles from the lower respiratory tract being the source of infection transmission from smallpox patients [8]. In fact, asymptomatic contacts have been documented to have variola in the oropharynx, but are not infectious and only a minority go on to develop smallpox [53]. This suggests that transmission of infection occurs preferentially with a lower respiratory infection, which would generate fine airborne infectious particles [8].

It should be noted that the theory of aerial convection was first proposed in England in 1881 following the observations around Fulham and that the data collected since then was in an era of rapidly declining incidence of disease, well after compulsory smallpox vaccination in the country in the mid-1800s. It is likely that prior to routine vaccination, transmission in the community was too widespread and intense to observe unusual patterns of aerial transmission from a single case to others. If there were many cases in a community, any incident cases would be attributed to close proximity transmission. It was only in the period of decline in the incidence of smallpox that the phenomenon became apparent because explanatory source cases in the community were largely absent. During the 100 years leading up to eradication, the low incidence of smallpox made it easier to observe unusual transmissions from single cases, and exclude close contact with a known case. There was, therefore, a limited window to collect data before smallpox became exceedingly rare in England [8]. Whilst the theory was debated and disputed, including the role of climatic wind conditions in the dispersion of smallpox by air, by 1904 more experts were in favour of aerial convection than against. However, by this time smallpox became too rare to collect ongoing outbreak data, and we are left only with the data from documented outbreaks between 1881 and 1971 [8]. Other than the systematic analysis of data attempted in England in the late 19th century, we are left with evidence from the individual outbreaks reviewed in this study.

In 1886, Sir George Buchannan addressed the Epidemiologic Society of England on the topic of aerial convection of smallpox: “We cannot get away from these facts; they are as definite as any known to epidemiology. They had already been ascertained by a multiplicity of careful and detailed observations, in many hospitals, in different epidemics, in London and the Provinces. Recent epidemics have now enabled the question to be tested afresh. That smallpox hospitals have had a deleterious influence in disseminating the disease in surrounding areas is now admitted, so there is no need for this aspect of the case to be argued further, but it is noteworthy that with no other disease has a similar influence been established. In this respect, smallpox stands alone, which proves that its infectivity is exceptional” [8]. The uniqueness of smallpox transmission in contrast to other infections is the striking point made. There are alternative explanations to cases occurring within the incubation period of theoretical exposure to a distantly located primary case. In the Purfleet examples, some experts felt that staff were visiting the communities onshore in secret, possibly carrying with them contaminated clothing or bedding. It was also postulated in Fulham and the rest of the English hospitals that the rings of infection around the buildings were due to movement of staff wearing contaminated clothing. Secondary aerosolization of virus from scabs or other bodily secretions on clothing is possible. The hospital staff, if immune, could conceivably carry fragments of scabs on their clothing which could infect susceptible community contacts whilst the staff themselves remained well. It was documented that smallpox particles are extraordinarily resistant to inactivation by drying (low humidity conditions) and if not exposed to direct sunlight, can remain within dust particles for long periods of time (54,55). If this is the case, isolated single cases could occur by re-aerosolisation of scab material on clothing or bedding. One study in 1957 showed that scabs from smallpox patients could contain the viable virus for 18 months, and for years if the scabs were kept in bottles [54]. In a later study in 1967, Wolff and Croon showed that in dried crusts from skin lesions of variola minor, the virus can remain viable for at least 13 years at room temperature [55].

## 5. Conclusions

In summary, the evidence from these outbreaks is supportive of aerial convection of smallpox at distances of more than a mile in some cases and is biologically plausible due to higher concentration of virus in the lower respiratory tract, environmental factors such as wind, and the low infectious dose. In addition, in many of the observed long-range transmissions, there was a temporal association between potential exposure to a known case and illness. It is possible, that some cases of smallpox were “super-spreaders” with much higher viral shedding than others. This has been seen with other viral respiratory pathogens such as SARS. If this is the case, super-spreaders could explain long-range transmission.

The theory of aerial convection arose in the period of decline in smallpox incidence in the UK, as the rarity of the disease made it possible to notice unusual transmissions in the absence of close contact. This small window of opportunity for studying aerial convection then rapidly closed, as smallpox became extremely rare in the UK by the early 20th century. This may, in part, explain the loss of this disease transmission theory from current infection control policy and practice, but potentially places smallpox in a different category from other known respiratory transmissible infections. In modern hospitals in high-income countries, negative pressure isolation rooms would reduce any risk of aerial convection. Other explanations for apparent aerial convection are possible, including missed chains of transmission, fomite transmission and secondary aerosolization of contaminated materials such as bed linen. Should smallpox re-emerge, awareness of the possibility of aerial convection is important, as it could inform planning for smallpox treatment facilities and protecting hospitals and surrounding communities.

## Figures and Tables

**Figure 1 viruses-12-00033-f001:**
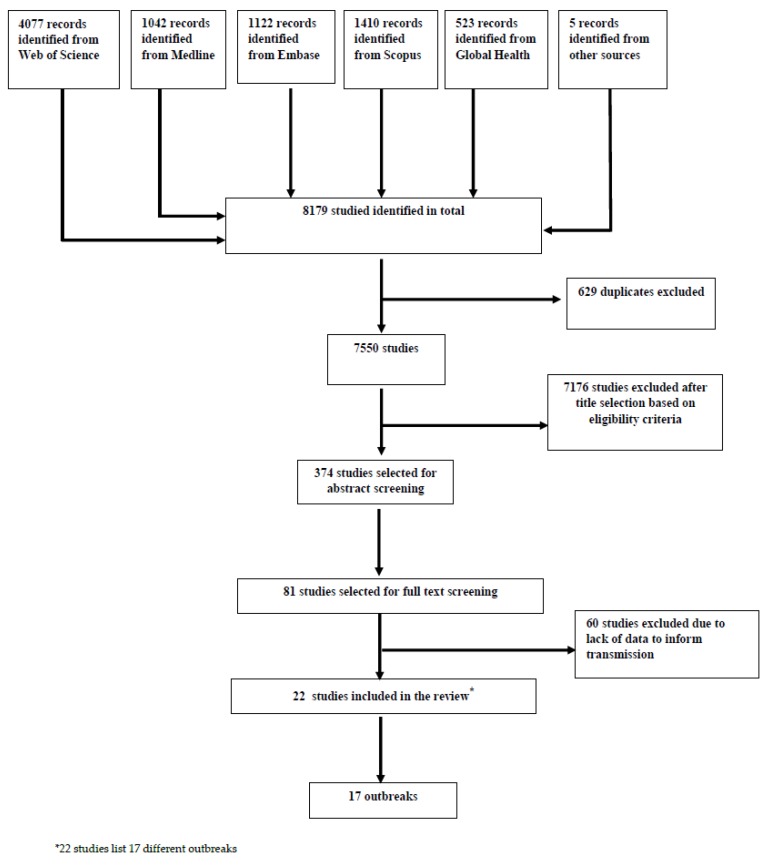
Search process for systematic review [34].

**Figure 2 viruses-12-00033-f002:**
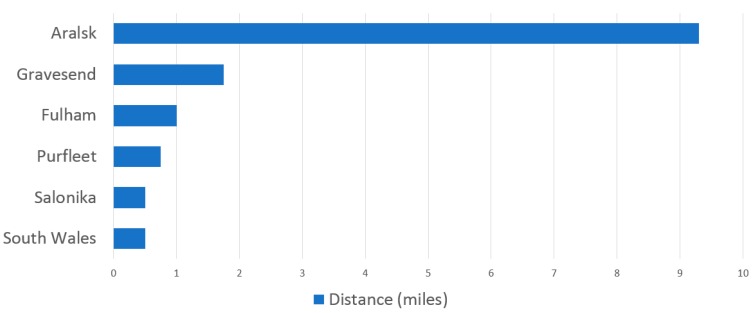
Maximum reported distance (miles) of aerial convection of smallpox in different outbreaks, where distance was quantified.

**Figure 3 viruses-12-00033-f003:**
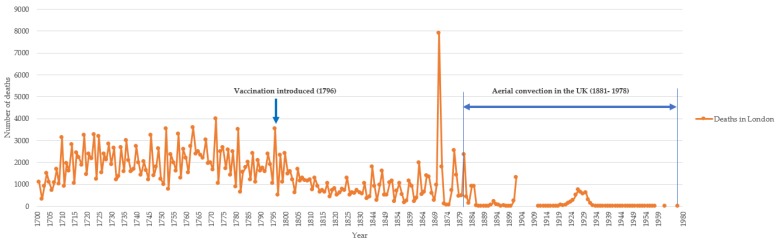
Number of smallpox deaths 1700–1980, London—period of observation of aerial convection shown in two-sided horizontal arrow.

**Table 1 viruses-12-00033-t001:** Outbreaks providing evidence of aerial transmission of smallpox.

Year of Outbreak	Reference (s)	Country/City	Effect of the Outbreak	Findings in Relation to the Evidentiary Threshold	Main Results
1881	[8,36]	Fulham, United Kingdom	A total of 62 cases of smallpox among the residents of Fulham, of whom 32 lived within a one-mile radius of the hospital, and 23 cases had no known exposure to smallpox cases.	Conclusive	Cases occurred in the community in an area immediately around smallpox hospitals, without known contact with smallpox cases or explanatory smallpox cases in the surrounding community. As supportive evidence, the density of smallpox cases around Fulham hospital fell away proportionally to distance. In 1881, smallpox hospitals were located in densely populated urban areas. That year, WH Power, the medical inspector, showed through a series of disease maps that there was an increasing incidence of smallpox in the city with increasing proximity to the hospital. This effect was observed for at least 1 mile. He investigated the movement of hospital staff and patients and concluded that contact with patients or hospital personnel could not explain the observation. Further, the disease map patterns were not concentrated near main traffic units to and from the hospital highlighting a possible difference in the mechanism of transmission. Power was the first person to attempt to quantify this and coined the term “aerial convection”. A report in 1886, “Statistics of Smallpox Relative to the Operations of Hospitals in the Metropolis” confirmed that the same phenomenon was observed with other smallpox hospitals in London and in the provinces.
1901–1902	[8]	Orsett Union (Purfleet), Essex, United Kingdom	Hospital ships were used to treat smallpox in London. In 1901-2, during an epidemic, the hospital ships were moored in the Thames river near the Kentish shore. The Orsett Union on the other side of the river suffered an epidemic. The community closest (3/4 mile) to the river, Purfleet, had 110 households, of which 44 were infected with smallpox, without known contact with cases. The next nearest community to the ships was West Thurrock, which suffered the next largest epidemic.	Conclusive	Smallpox cases occurred in the nearby community onshore, closest to the hospital ship. Aerial convection was the only explanation, as there was no documented mixing of people from the ship with the onshore community. Some speculate there were clandestine visits by the crew to shore, but only one such instance was reported. Further supportive evidence is that the same phenomenon of epidemics in Purfleet following mooring of the hospital ships was observed in three consecutive epidemics.
1917	[37]	Salonika, Greece	The index case (an infected seaman) infected 1 person, who was half a mile away from the hospital in Greece where he was admitted, within the incubation period. As a result, there were 20 further cases and several deaths.	Conclusive	The index case was isolated in a tented camp half a mile away from any habitation. No other source of infection was identified and there were no other cases of smallpox in the nearby community. The likely mode of transmission was through aerial convection of virus particles. The flow conditions could also likely be more conducive for particulate travel due to the open ‘sandy’ terrain between cases with appropriate wind conditions.
1920	[38]	Glasgow	An epidemic in Glasgow occurred where people in the vicinity of the treating hospital became ill. A physician reported that a case of smallpox was admitted to the “smallpox compound of a hospital in a distant part of the city at a time when no other cases were occurring, and this was the only case of smallpox in that hospital.” After the incubation period, cases occurred in the scarlet fever ward, which overlooked the smallpox compound.	Partially conclusive	During this epidemic, subsequent cases occurred in the vicinity of the hospital which was treating smallpox cases. A clinician specifically recalls cases occurring in an overlooking ward across a spatial separation of much greater than 2 m from the smallpox ward (where a single smallpox patient was treated) after the incubation period.
1938	[37]	Gravesend, United Kingdom	Smallpox occurred in 6 cases in the UK, who had not been in contact with the index patient, and were located at a distance of ¼ to 1 ¾ miles from the index case. No other possible source of infection was identified.	Conclusive	The secondary cases lived at a distance ranging from a quarter of a mile to one and three-quarters of a mile from the hospital where a smallpox case was being treated, and no other source of infection was identified. There was no smallpox in the community at the time. The spread might have been an unusually dispersive type or associated with wind or other weather conditions.
1947	[39]	New York, USA	The index case travelled from Mexico to New York and was admitted to the ground floor of a hospital, with a subsequent outbreak consisting of 12 cases, 9 originating in New York City and 3 in Millbrook, N. Y. Patients who did not have contact with the index case, including one patient on the 7th floor, became infected.	Conclusive	The index case, a 47-year-old merchant who developed a rash and got admitted to Bellevue Hospital where he remained until March. He was transferred to Willard Parker Hospital, the communicable disease hospital in Manhattan. Among the patients who were in Willard Parker Hospital at the same time as the secondary cases were a male age 27 with mumps, and a 22 months old girl suffering from croup, a 2 and half-year-old boy suffering from whooping cough. All of them had been in the same building: the index case and the baby on the ground floor and the 47-year-old male on the seventh floor. Three men, aged 43, 57, and 60, all patients at Bellevue Hospital when the index case was there, got infected. A 4-year-old boy who got discharged on the day the index case died, developed rash later. He was the source of 3 other cases—a nun, aged 62, a 5-year-old boy, and a 2-year-old girl. There is a probability of airborne transmission of smallpox from the index case to 9 secondary cases residing in the same hospital building but spatially separated, including on different floors. There was no documented direct contact between the cases, with one secondary case six floors above the index case.
1947	[40]	Germany	The index case caused an outbreak of smallpox infection in 18 additional persons, both in the hospital and surrounding community.	Partially conclusive	The Army hospital in Wiesbaden, Germany received a patient with a presumptive diagnosis of smallpox. Although the patient was placed in strict isolation, there were secondary cases of smallpox in hospital patients and 7 people in the surrounding community, who did not have direct contact. The hospital cases were patients admitted in widely scattered parts of the hospital and had no known contact with the index case. There were no other smallpox cases in this area prior to the outbreak. Multiple possible means of transmission were postulated, including transfer by fomites, through laundry procedures and bed linen, contaminated blood collection equipment and aerial convection. The possibility of virus-laden dust moving in natural air currents was considered in explaining cases arising in the surrounding community and across large distances within the hospital.
1961	[41]	Monschau, Germany	The index patient was a 9-year-old girl admitted to a hospital in Germany. On that ward, 10 persons, none of whom had direct face-to-face contact with the source case, developed smallpox.	Conclusive	Infection appeared to have spread within the hospital through aerial transmission over a considerable distance along the common corridor and suggesting that air flowed from the isolation unit to the neighbouring ward.
1962	[42]	Simmerath, Northrhine-Westphalia, Germany	The index patient was a 6-year-old child admitted in the hospital who subsequently infected 10 more persons	Conclusive	At that time, a heavy wind was affecting the gable-end of the building, which has given rise to a draught in the corridor and infected people at the opposite end of the corridor. A wind tunnel effect was observed in this case, which was suggestive of aerial transmission.
1962	[43]	England, United Kingdom	There was a total of 62 infected cases, out of which no explanation of the source could be explained in 4 cases.	Conclusive	Four districts were involved: Woolwich Met. B., Hornchurch U.D. Rhondda M.B. and Penybont R.D. In the whole series of 62 cases, all but four could be attributed to clear direct sources of infection. In three of the four entirely unexplained cases, there was no evidence of contact or possible sources of infection under surveillance within 150 miles. Regarding the Penybont incident, there were three cases of smallpox, a few miles in Llantrisant not under surveillance, with no contact either direct or indirect with the Glanrhyd Mental Hospital, Bridgend, where a smallpox case was treated. This indicates a possible spread of infection from smallpox hospitals, presumably in the form of airborne infection.
1962	[26]	South Wales, United Kingdom	There were a total of 26 patients out of which 5 died.	Conclusive	This was related to a cluster of outbreaks which occurred in England in the same year [43]. The index case, a Pakistani male was infected with smallpox and was admitted to an isolation ward in a hospital in Cardiff on January 15th. He was ill with benign confluent smallpox and was discharged from hospital on March 6, 1962. In this outbreak, a woman died during childbirth of probable smallpox on February 9th. It was found out that this woman’s house was about half a mile from Penrhys Hospital, where the Pakistani patient from Cardiff was a patient. There were no other cases of smallpox in the region. The woman/s house was about half a mile from Penrhys Hospital, where the patient from Cardiff was admitted from January 16 until March 6, raising the possibility of aerial convection.
1962	[26]	South Wales, United Kingdom	20 smallpox cases and 12 deaths were reported in this outbreak	Partially conclusive	This is the continuation of the cluster of outbreaks in England [43]. A 75-year-old woman admitted in the Heddfan hospital ward for many months and probably died of smallpox infection. Many patients got infected in the Glanrhyd Hospital and this hospital is situated about half a mile from Heddfan Hospital, where the index patient was admitted. There were no other cases of smallpox in the region. There was no evidence of the transfer of virus via persons or items from inside Heddfan to the outside and there was no evidence of poor infection control by the staff members. This suggests the possibility of aerial conveyance of smallpox virus from the infected woman in the Glanrhyd hospital to the patients in the Heddfan hospital.
1963	[44,45,46,47]	Sweden	A 24-year-old sailor flew from Australia to Zurich with seven stops and contracted smallpox either on the aircraft or at an airport terminal. He eventually caused 24 secondary cases and 4 deaths.	Partially Conclusive	The index case had not been in contact with any known sources of infection. The index case was attributed to in-transit exposure either at the airport terminal or the aeroplane and this might be indicative of airborne transmission.
1967	[48]	Kuwait	A total of 41 cases and 19 deaths were recorded. Of 41 cases, at least 32 had a known close contact with unrecognized smallpox patients admitted to the hospitals.	Partially conclusive	The first case occurred in a Pakistani woman who arrived in Kuwait from Karachi by air and was admitted to the Fever hospital after she became ill. An outbreak of smallpox was, however, not suspected until a second-generation case was diagnosed. The first case caused 3 infections which subsequently caused an outbreak in the Fever hospital, Kuwait. In the third generation, there were a total of 10 cases and at least 8 of them had hospital contacts. However, for 2 of these patients, the source of infection could not be traced. One unrecognized third-generation case was transferred from the Fever Hospital to the Sabah Hospital which resulted in further hospital-associated cases in the Sabah Hospital, 25 cases occurred in the fourth generation and out of them, 4 cases did not have a definite history of contact with patients either in the Fever Hospital or in the Sabah Hospital. Three cases had previously been admitted to the Fever Hospital but had been discharged more than one incubation period previously. No family contact was involved in these four cases. The unknown source of infection in these cases may indicate aerial or airborne transmission of smallpox.
1970	[9,10,11,17,41,42,49]	Meschede, Germany	An index case in hospital infected other patients on different floors of the building, 21 cases of smallpox occurred, with 4 deaths.	Conclusive	A smoke test of the air movement in the building under similar atmospheric conditions strongly suggested aerial convection. The pattern of airflow coincided closely with the distribution of cases within the hospital. Further, through examining smoke escaping out of the window from the index patients’ room, this provided evidence of re-entry of air into the building from windows above where cases were present. The lower humidity conditions (reported in the paper) were also likely to have led to increased airborne dispersion.
1971	[50]	Aralsk, Kazakhstan (Soviet Union)	A crew member on the Lev Berg ship contracted smallpox during a voyage of the ship, which passed 15km from Vozrozhdeniye Island, a Soviet biological weapons testing site. A total of 11 secondary cases of smallpox resulted from this case.	Conclusive	The index case was a young fisheries expert who was on a biological research ship called the Lev Berg, sailing from Aralsk to multiple locations on the Aral Sea. She contracted smallpox at the end of July 1971. Official reports claim she got off the ship at several stops, but in an interview she denied it. Official policy was that only male crew were allowed to disembark. During the voyage, she spent a large amount of time outdoors on the deck, collecting samples. Dr Pyotr Burgasov, chief sanitary physician, said in an interview in the Russian press in 2001 that weaponized smallpox was being tested on Vozrozhdeniye Island (a Soviet biological weapons testing site) at the time. It seems likely this was the source of infection, with virus travelling at least 15 km to infect the index case. Whether this was the result of deliberate dispersal of the virus through smallpox “bomb” being tested, or accidental dispersion from the lab, is unknown, although Dr Burgasov is quoted as saying “The smallpox formulation—400 g of which was exploded on the island—“got her” and she became infected.”
1978	[27,28]	Birmingham, United Kingdom	A fatal case of smallpox occurred in a photographer working on the first floor of the medical school above the department of microbiology (where smallpox research was being done). She had never visited this laboratory. There were no other cases among people on the first floor.	Conclusive	Some of the smallpox laboratory work on the ground floor was carried out inside a safety cabinet, using unsafe procedures. The reports also suggest that “the opening and closing of the smallpox room door and the passage in and out by whoever was conducting work on the virus would have created the opportunity for any airborne virus to escape into the animal pox room”. The service ducts in the animal pox room and the smallpox room both had gaps which could allow the leakage of viruses. There also might have been a possibility of aerosol transmission from the ground floor to the first floor either through the air conditioning system or through an open window.

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
