# Peer review of "Evidence of Long-Distance Aerial Convection of Variola Virus and Implications for Disease Control"

_viruses, 2019, doi:10.3390/v12010033_

Round 1

Reviewer 1 Report

This is a fascinating paper that uses a systematic review of historical smallpox data to determine the likelihood of aerial convection of variola virus. Despite my initial scepticism the authors actually present a pretty convincing case for aerial convection and I have no substantive concerns. The systematic review appears to be correctly performed.

I only have two minor issues:

The title is technically incorrect. It is not smallpox that is transmitted but the causative variola virus. So, virus needs to be noted in the title. The link between the systematic review of global smallpox outbreaks and the epidemiology of smallpox in London is not totally clear to me. Can this link be clarified and made more explicit.

Author Response

(1) We modified the title to be “Evidence of long-distance aerial convection of variola virus and implications for disease control” and used the word variola instead of smallpox in relevant sections throughout the document.

(2) We revised the paper to expand on the rationale by revising the aims and methods as follows:

a) “The aim of this paper was to study the data and epidemiology of transmission of variola by aerial convection, and examine the hypothesis that aerial convection can only be observed when incidence of smallpox is sufficiently low to rule out other sources of infection.” (page 2, lines 84-86)

b) “We used the systematic review methodology to identify data and evidence for aerial convection, and then analysed incidence data, documented distances of transmission, and the period of observation of aerial convection in relation to the epidemiology of smallpox deaths.” (page 2, lines 90-93).

c) ”In order to clearly demonstrate the window during which aerial convection was observed relative to the epidemiology of disease, we created the timeline of all identified observations of aerial convection that was plotted over the number of smallpox deaths using available data in London from 1700-1980. London was selected because there was better data for deaths (than cases) than global data or other countries, and the largest number of observations of aerial convection were from England.

We collected the data of the annual number of deaths from smallpox in London from 1700 to 1902(23). Due to missing data on deaths in London between 1903 and 1958, we collected the available data to estimate data for London, including ①the number of deaths from smallpox in England and Wales from 1911 to 1919(24), ②population in London, England and Wales in 1911(26), ③ number of cases and case fatality rate in the UK from 1920-1958(25), and ④ the number of deaths in London between 1959 and 1980(31-33).” (Please see line 95-105, page 3)

Reviewer 2 Report

I enjoyed reading this manuscript. It was well-written and easy to follow. I was only vaguely aware of the concept of "aerial convection" prior to reading this article. I do think this study adds important concepts to our understanding of smallpox transmission.

I have just a couple minor points/suggestions:

line 33: The authors comment about the potential threat of smallpox re-emergence due to synthetic biology. I would also add that the absence of mass vaccination nowadays also plays a role in this threat.

line 66 and line 114: I would like the authors to be more clear about the greater than 2 m criteria. Is this a previously agreed upon distance in the literature or a criteria that the authors of this study devised?  [as a side-note: this distance seems rather close to be used in a discussion about aerial convection; some of the transmission events occurred over very large distances with no known contact between individuals - this seems more in line with such a theory. I think the average reader would think a distance of several meters is still direct contact. Perhaps the authors can address this specifically.]

Author Response

(1) We have revised the statement to: “However, smallpox poses a public health threat of re-emergence due to the absence of mass vaccination, waning immunity and advances in synthetic biology, which make synthesis of variola possible(3, 4).” Please see lines 34-35, page 1.

(2) We revised the paper as follows:

a) “The aim of this paper was to study the data and epidemiology of transmission of variola by aerial convection, and examine the hypothesis that aerial convection can only be observed when incidence of smallpox is sufficiently low to rule out other sources of infection.” (page 2, lines 84-86)

b) “We used the systematic review methodology to identify data and evidence for aerial convection, and then analysed incidence data, documented distances of transmission, and the period of observation of aerial convection in relation to the epidemiology of smallpox deaths.” (page 2, lines 90-93).

c) “In addition to direct respiratory transmission from person to person within 1-2 meters of spatial separation, more distant transmission has been described. Whilst it is well-established that airborne infection can occur(8, 19), the spread of smallpox by means of “aerial convection” is less well understood. Aerial convection refers to transmission over a substantial distance, (greater than expected during direct person to person respiratory transmission of 1-2 metres and possibly aided by wind or air currents) a concept accepted by many epidemiologists. In recognition of this, Ministry of Health regulations in Britain in the 1940s stipulated that smallpox hospitals should be “at least a quarter of a mile from another institution or a population of 200, and at least a mile from a population of 600.” (page 2, lines 68-72)

“Then we plotted all identified observations of aerial convection over this timeline, to show the relationship of this observation period of aerial convection to the epidemiology of smallpox and the waning incidence of smallpox.” (page 3, lines 114-116)

Reviewer 3 Report

Title: Evidence of long-distance aerial convection of smallpox and implications for disease control

The manuscript describes evidence from the literature for significant spread of smallpox by aerial convection. The authors are aware of, and describe, the difficulties in studying aerial convection, namely the short window of opportunity for the studies between introduction of widespread vaccination and the eradication of the disease; and the relative paucity of definitive data.

Unfortunately the authors do not discuss a significant alternative explanation for the long (>2m) distance spread of smallpox, namely transmission by contact with infectious people who are unrecognized by the medical community.  This alternative possibility was recognized at the time, and held to be highly plausible by one of the most highly respected experts, C. W. Dixon, in his seminal book “Smallpox” (1962, Publisher J. & A. Churchill Ltd, London). On Page 372 Dixon states:

 “it is surprising how well the evidence could be made to fit certain outbreaks. Smallpox occurred in nearby houses shortly after the admission of early cases, the occurrence of smallpox in the path of the prevailing wind, the experience at Purfleet (Thresh, 1902 [Thresh, J. The Hospital Ships Of The Metropolitan Asylums Board And The Dissemination Of Smallpox. The Lancet 159, 495-498.  1902]), with the apparent transfer of infection across the river, a distance of nearly half a mile from the floating river hospital… … the pattern, however, did not always fit, and Hope (1905) [ Hope, E. W. (1905). “Observations by the Medical Officer of Health on the Report of Dr. Reece on Smallpox in Liverpool.” – I have not been able to find this reference, only Dixon’s description of it], in Liverpool, showed that when a smallpox outbreak occurred in a town, an increased incidence occurred around a hospital if it was not admitting smallpox at all.”

Further down he states:

“It is almost certain that the safeguards against the communication of hospital staff, particularly porters and others, with the outside world, is far less secure than most medical superintendents of smallpox hospital imagine. It is very unlikely that questioning staff ever brings to light some of the irregularities of practice which have occurred, and which could account for spread of infection”

Dixon does not dismiss aerial convection, but he clearly opines that unrecognized direct transmission by person-to-person contact is a likely alternative explanation in many cases. He is not alone in this; Millard (Millard, C. K. Does vaccination help to spread smallpox? Lancet, 104-107. 1951) speculated that vaccination facilitated the spread of smallpox, because vaccinated people would sometimes develop a mild or unrecognized infection that would not be reported to medical practitioners, but which could be capable of transmitting infection to non-vaccinees.

Outwardly, the Aral sea event, the 1978 Birmingham outbreak, and the Meschede hospital outbreak appear to be very strong evidence for aerial convection.  The Aral sea event is clear, however it almost certainly involved a large quantity of well prepared virus stock.  Material lofting from a patient or hospital would be much lower titre, and subject to a very high dilution effect proportional to distance from the source. So the Aral sea event is important for response to a deliberate release event, but not necessarily to responses to subsequent foci.

The events around the 1978 Birmingham outbreak are not at all clear, despite there having been a Government inquiry. I do not think the expert community then accepted that airborne transmission between rooms was responsible, nor do I think they do now.  It is not clear how Mrs. Parker came to be infected, but it is clear that she could have come into contact with the virus by a route that had nothing to do with aerial convection, and that the inquiry could plausibly have been unable to establish this.

The Meschede hospital outbreak is also not definitive. The smoke pattern/case distribution analysis in Wehrle et al. (1970), as described in Chapter 4 of “Smallpox and its Eradication Fenner et al. (World Health Organization Geneva, 1988).” could be interpreted either way, as supporting or not supporting aerial convection. The important feature that is missed is that the Meschede hospital was a hospital in which an unexpected case of smallpox was cared for. Aerial convection is a hypothesis to explain the subsequent cases that occurred, but the null hypothesis, that spread was by a different route, was not disproven.

Major Correction

The authors need to explain that aerial convection is a hypothesis for which there is clear evidence in the form of the Aral sea event. Although there is further evidence from other events, this is less clear, and the risk posed by person-to-person transmission from unrecognized infection (e.g. in people who may have received prior vaccination), should be considered by emergency planners, as well as the risk posed by aerial convection.

Minor corrections

Line 30  “Orthopoxvirus family”  Poxvirus is the family, Orthopoxvirus is the genus

Line 47, I don’t think these studies show that a single pfu is enough to cause infection with variola. The Westwood paper demonstrates this for rabbitpox, but not variola.  Should say rather “orthopoxvirus infections can be produced by a single plaque forming unit (PFU) of virus carried in respirable particles”

Line 73 “have the potential to remain suspended in the air for long periods of time”  How long?  My suspicion is hours, but the statement could mean anything.  It needs a limit, or to state that it is supposition.

Line 74 “larger number of susceptible individuals to infections at a greater distance from the source” this is a misleading statement, because as the aerosolized material disperses it becomes less likely to contact a person, even though the number of people in the dispersal area is greater than in the pre-dispersal area.  Should actually say “larger number of susceptible individuals to potential infections at a greater distance from the source”  it may seem trivial, but it’s actually quite important.

Line 140 “or where the outbreak was occurred in the presence” remove “was”

Line 221 “In fact, in the dried crusts from skin lesions, the virus can remain viable for years at room temperature”  This absolutely requires a primary reference, otherwise it is urban myth.

Author Response

Major correction

We have revised the abstract and conclusion to acknowledge the possibility of other explanations for aerial convection, and softened the language throughout (such as using “supportive” rather than “conclusive”). We have also added that modern hospitals with negative pressure rooms would minimise any risk.

See lines 22-23 (abstract) and lines 274-280 (discussion):

“In modern hospitals in high income countries, negative pressure isolation rooms would reduce any risk of aerial convection. Other explanations for apparent aerial convection are possible, including missed chains of transmission, fomite transmission and secondary aerosolization of contaminated materials such as bed linen. Should smallpox re-emerge, awareness of the possibility of aerial convection is important, as it could inform planning for smallpox treatment facilities and protecting hospitals and surrounding communities.”

Minor corrections

(1) We have revised it to: “poxvirus family”. Please see Line 32.

(2) We have revised the statement to: “the orthopoxvirus infection can be produced by a single plaque forming unit (PFU) of variola virus carried in respirable particles”. Please see Line 50-51.

(3) We have revised the statement to: “Aerosolised particles have the potential to remain suspended in the air for hours and can expose a larger number of susceptible individuals to potential infections at a greater distance from the source (21)”. Please see Line 76-77.

(4) We have removed “was” as suggested. Please see Line 151.

(5) We have revised the statement and added the primary reference for the statement: “In fact, viable virus can be found in large quantities in the dried crusts from skin lesions (51). Please see line 229-230.

Round 2

Reviewer 3 Report

Major Correction:

The authors have inserted minor caveats that missed chains of transmission are an alternative explanation for the long range transmission. This does not give missed chains of transmission sufficient weight. As far as I can see, the intent of the manuscript is to highlight that unusual mechanism(s) of transmission occur with smallpox, different from what one might expect with outbreaks of diseases such as influenza or SARS; and thus of importance to emergency planners. This is important and laudable. However, by concentrating on aerial convection to the effective exclusion of missed chains of transmission, the authors are failing to inform readers of the potential importance of missed chains of transmission. Given that smallpox vaccinees are susceptible to infection which can lead to extremely mild disease (for example one or two lesions, or indeed many lesions but potentially with none visible outside clothing), unrecognized chains of transmission are a highly plausible mechanism for all but the Aral sea event. The possibility that quarantine and/or social exclusion (with concomitant loss of income); or simple ignorance or non-acceptance of the risks, could have influenced vaccinated people to not report mild infection seems to me a simpler explanation for most cases of unexplained transmission. If I am right, then to publish this manuscript in its current form would be counterproductive. I agree entirely that long-range transmission of smallpox is a difference to what we observe with many serious virus diseases, and this is the novel or interesting feature of this subject. But the manuscript must give more prominence to unrecognized transmission chains. It is acceptable to conclude that aerial convection can occur – the Aral sea event is convincing. It is thus necessary to extrapolate the Aral sea event to other events, but the conclusion there is that aerial convection and transmission by vaccinees should both be considered by emergency planners. This requires a major revision, not a minor caveat such as the authors have given.

Dixon's seminal book has been out of print for decades, but it is surely available from national or university libraries around the world. Reading Dixon it is clear that he was unconvinced by aerial convection, because he recognized holes in systems to prevent transmission by hospital workers and others associated with hospitals.

Also, anecdote is a frustrating feature of the history of late 20th century smallpox transmission. Somewhat unfortunately, both the 1978 Birmingham outbreak and the Meschede hospital outbreak are the subject of cryptic anecdotes about features of which the enquiries were not made aware. Although one cannot place weight on them, it cautions us to look carefully at the alternative explanations in addition to aerial convection.

I would like to see this subject brought to the attention of the community, but cannot accept it excluding what I see as a likely explanation for some or many of the apparent long range transmission events.

Minor Corrections:

In line 52-53, the new version still specifies variola virus. “variola” should be removed from the phrase “orthopoxvirus infection can be produced by a single plaque forming unit (PFU) of variola virus carried in respirable particles”

In Line 238 a review (rather than primary) reference has been inserted. However the statement it is intended to support has been changed. The original statement (that I questioned) pertained to long-term viability of virus in crusts at room temperature. The statement now pertains simply to the quantity of virus in crusts, which will certainly be high at the time of crust development. As such, the new statement is a non-sequitur that has no relevance to aerial convection, especially as crust material will be large particle size, and the authors present evidence that large particles are highly unlikely to be capable of aerial convection. The new statement should be removed. As the references do not support the old statement, the old statement should not be re-inserted.

Author Response

Major corrections:

We have revised the paper extensively to include a more detailed discussion of secondary aerosolization of dried scab material and missed chains of transmission. In Discussion, we added the following statements.

Lines 225-234: “The examples of transmission from one floor to another or one building to another, presumably by air currents, are more easily explained, as distances were shorter and supported by the smoke experiments at Meschede, Germany (9, 10, 49). In the last documented case of smallpox in the world, the Birmingham case, we can be fairly certain the patient was infected from a virus in the laboratory. Case ascertainment was high at that time, the location was the UK, which had long since eliminated smallpox, so it is likely the source of infection was an aerosolised virus through air-conditioning dusts or an open window.. More recently, transmission of SARS in the Amoy Gardens building, where aerosolised faecal material spread from floor to floor through plumbing and open bathroom grates, but also from open windows to adjacent buildings, demonstrated that air currents can carry virus particles from one building to another(52).”

Lines 239-242: “Another possibility in the apparently long-range transmissions is that these were exposed to missed mild or vaccine-modified cases. However, it should be noted in the case in Greece, that the secondary case occurred within the incubation period of the index case being symptomatic (37).”

Lines 281-295: “There are alternative explanations to cases occurring within the incubation period of theoretical exposure to a distantly located primary case. In the Purfleet examples, some experts felt that staff were visiting the communities on shore in secret, possibly carrying with them contaminated clothing or bedding. It was also postulated in Fulham and the rest of the English hospitals that the rings of infection around the buildings were due to movement of staff wearing contaminated clothing. Secondary aerosolization of virus from scabs or other bodily secretions on clothing is possible. The hospital staff, if immune, could conceivably carry fragments of scabs on their clothing which could infect susceptible community contacts whilst the staff themselves remained well.  It was documented that smallpox particles are extraordinarily resistant to inactivation by drying (low humidity conditions) and if not exposed to direct sunlight, can remain within dust particles for long periods of time (54,55). If this is the case, isolated single cases could occur by re-aerosolisation of scab material on clothing or bedding. One study in 1957 showed that scabs from smallpox patients could contain viable virus for 18 months, and for years if the scabs were kept in bottles(54). In a later study in 1967, Wolff and Croon showed that in dried crusts from skin lesions of variola minor, the virus can remain viable for at least 13 years at room temperature (55)”.

Also in Conclusions on lines 300-304, we added the following statement: “In addition, in many of the observed long range transmissions, there was a temporal association between potential exposure to a known case and illness. It is possible that some cases of smallpox were “super-spreaders” with much higher viral shedding than others. This has been seen with other viral respiratory pathogens such as SARS. If this is the case, super-spreaders could explain long-range transmission.”

Minor corrections:

(1) Line 51: We removed “variola” from the phrase as suggested.

(2) We added two primary references “54. MacCallum F, McDonald J. Survival of variola virus in raw cotton. Bulletin of the World Health Organization. 1957;16(2):247” and “55. Wolff HL, Croon J. The survival of smallpox virus (variola minor) in natural circumstances. Bulletin of the World Health Organization. 1968;38(3):492” for survival of variola in dried scabs for long periods. The modified statement (lines 281-295) has been moved to support the possibility of secondary aerosolization of scab material  from clothing or bedding to explain isolated cases without known contact.

We have also made some minor edits for clarity.